# Digital Transformation Will Change Medical Education and Rehabilitation in Spine Surgery

**DOI:** 10.3390/medicina58040508

**Published:** 2022-04-02

**Authors:** Tadatsugu Morimoto, Hirohito Hirata, Masaya Ueno, Norio Fukumori, Tatsuya Sakai, Maki Sugimoto, Takaomi Kobayashi, Masatsugu Tsukamoto, Tomohito Yoshihara, Yu Toda, Yasutomo Oda, Koji Otani, Masaaki Mawatari

**Affiliations:** 1Department of Orthopaedic Surgery, Faculty of Medicine, Saga University, Saga 849-8501, Japan; h.hirata.saga@gmail.com (H.H.); s2001007@yahoo.co.jp (M.U.); o907471613b@gmail.com (T.S.); takaomi_920@yahoo.co.jp (T.K.); masa2goo99@yahoo.co.jp (M.T.); tomohito4113@yahoo.co.jp (T.Y.); darapon414@gmail.com (Y.T.); mawatam@cc.saga-u.ac.jp (M.M.); 2Education and Research Center for Community Medicine, Faculty of Medicine, Saga University, Saga 849-8501, Japan; norio.fukumori@gmail.com (N.F.); oday@cc.saga-u.ac.jp (Y.O.); 3Innovation Lab, Teikyo University Okinaga Research Institute, Tokyo 173-8605, Japan; sgmt@med.teikyo-u.ac.jp; 4Deptartment of Orthopaedic Surgery, Fukushima Medical University, Fukushima 960-1295, Japan; kojiotani1964@gmail.com

**Keywords:** medical education, hologram, augmented reality, action camera, extended reality, mixed reality, virtual reality, navigation, spine surgery

## Abstract

The concept of minimally invasive spine therapy (MIST) has been proposed as a treatment strategy to reduce the need for overall patient care, including not only minimally invasive spine surgery (MISS) but also conservative treatment and rehabilitation. To maximize the effectiveness of patient care in spine surgery, the educational needs of medical students, residents, and patient rehabilitation can be enhanced by digital transformation (DX), including virtual reality (VR), augmented reality (AR), mixed reality (MR), and extended reality (XR), three-dimensional (3D) medical images and holograms; wearable sensors, high-performance video cameras, fifth-generation wireless system (5G) and wireless fidelity (Wi-Fi), artificial intelligence, and head-mounted displays (HMDs). Furthermore, to comply with the guidelines for social distancing due to the unexpected COVID-19 pandemic, the use of DX to maintain healthcare and education is becoming more innovative than ever before. In medical education, with the evolution of science and technology, it has become mandatory to provide a highly interactive educational environment and experience using DX technology for residents and medical students, known as digital natives. This study describes an approach to pre- and intraoperative medical education and postoperative rehabilitation using DX in the field of spine surgery that was implemented during the COVID-19 pandemic and will be utilized thereafter.

## 1. Introduction

In recent years, the concept of minimally invasive spine therapy (MIST) has been proposed as a treatment strategy to reduce the need for overall patient care, including not only minimally invasive spine surgery (MISS) but also conservative treatment and rehabilitation. This is because, while MISS has many advantages, there are limits to how much surgery alone can be used to maximize its effectiveness. Another problem specific to MISS is the narrow surgical field, which requires a very specific understanding of the anatomy related to MISS, as well as the lack of accurate spatial awareness during surgery, which can hinder education and the acquisition of surgical skills, especially in residents and medical students [1,2]. To overcome this problem and maximize the effectiveness of patient care, there has also been a focus on improving the education of medical students, residents, and patients through digital transformation (DX), including virtual reality (VR), augmented reality (AR), mixed reality (MR), and extended reality (XR), three-dimensional (3D) medical images and holograms, wearable sensors, high-performance video cameras, fifth-generation wireless system (5G) and wireless fidelity (Wi-Fi), artificial intelligence, and head-mounted displays (HMDs) [3]. VR is defined as “an immersive, completely artificial computer-simulated image and environment with real-time interaction” [4]. In spine medicine, it is frequently used for medical education, surgical simulation, and planning. AR is defined as “the concept of digitally superimposing a virtual object on a physical object in real space, allowing an individual to manipulate both simultaneously” [5]. MR, a hybrid of AR and VR, is the result of blending the physical world with the digital world [6,7]. XR is the collective name for VR, AR, and MR.

The unexpected onset of the COVID-19 pandemic has resulted in the widespread disruption of medical education and professional training for residents and medical students worldwide, with a reduction in elective surgeries and switching to conservative approaches, as well as restrictions on physical attendance at workshops and conferences, whenever possible [8]. As a result, to adhere to social distancing guidelines, the use of digital technology to maintain medical treatment and education is being more rapidly and innovatively implemented than ever before. Among DX modalities, the introduction of XR is rapidly progressing in the medical education field [8]. XR technology is computer-based and, therefore, allows the performance of learning activities that would not be possible in the real world. In addition, as expressed over a decade ago, “today’s students are no longer the people our educational system was designed to teach” [9]. Therefore, medical education must be adapted to respond to changes in students and society, to the evolution of science and technology, and to new educational needs and requirements. It has become essential in medical education to create a highly interactive and immersive educational environment and experience using immersive technology together with XR approaches for resident doctors and medical students, known as digital natives.

This study describes an approach to pre-, intra-, and postoperative medical education and rehabilitation using DX that was implemented during the COVID-19 pandemic and will be utilized thereafter.

## 2. Preoperative Medical Education: 3D Virtual Dissection Education for Medical Students

Anatomy is one of the most essential elements in medicine and surgery education. In many educational institutions around the world, the use of cadaver materials has given way to using images and virtual learning [10] because of the high educational effectiveness, reduced cost and time requirements, lack of availability of donated bodies, and risk of infection from cadavers [3,11]. In the field of spinal surgery, the anatomy is complex, and important nerves and blood vessels are in close proximity to each other, underscoring the importance of understanding the anatomy. Anatomical education using digital technology is useful for understanding and consolidating knowledge regarding this complex anatomy [10]. Our institution started to combine traditional gross dissection with 3D virtual dissection (Anatomage Table™; Anatomage, Inc., San Clara, CA, USA) in medical education in 2021.

### 2.1. Learning Anatomy of Medical Students

Practicing anatomy with cadaveric dissection has been considered useful not only for acquiring knowledge of anatomy but also for establishing professionalism among medical practitioners, human values, and ethics [12]. Although the practice of learning anatomy using cadavers has been traditionally conducted since the 17th century, there are advantages and disadvantages to this approach. The advantage is that the gross anatomy of each organ can be learned from a bird’s eye view by observing the actual human body directly [13]. The disadvantage is that gross anatomy practices with cadavers are very time-consuming. Indeed, Yaginuma et al. found in their study that the average time allotted for gross anatomy practice was 125 h, and 80% of medical schools stated that their students could not complete the work within the practice time [14]. Learning anatomy with cadavers takes a long time; thus, medical students cannot repeatedly receive the same training. Therefore, while students’ understanding of human anatomy does improve during the period of anatomy practice, the only way to study anatomy subsequently is to use textbooks or online resources (two-dimensional (2D) study).

### 2.2. Introduction of Digital Technology into Medical Education

To address these issues associated with anatomical learning, various learning devices have been developed and introduced into medical education. Furthermore, plastination and 3D printing digital models have been created. Although these models have slightly less detail than 2D study, they can be easily used to understand 3D structures and have been suggested to have positive learning effects [13,15]. Recently, VR has been used to project part or all of a human body into a 3D virtual space, whereas AR projects a 3D hologram superimposed onto a real space [13,16,17,18,19,20,21,22]. Medical education in colleges around the world has been dramatically affected by the onset of the COVID-19 pandemic [13], and face-to-face lectures, macroscopic anatomy training, and clinical clerkship training have not been available for some time. This environment has fostered the acceptance of the introduction of VR and AR technologies into anatomy education.

The Anatomage Table™, a virtual dissection table based on 3D visualization technology, can help medical students and residents learn about anatomy with the following advantages: (1) the table is easy to use and cover anatomy from the surface to the innermost depths of the human body; (2) it includes tissue images of each organ, allowing students to learn about gross and microdissection; (3) it can be used repeatedly, thus reducing the time and space costs of studying anatomy; (4) it includes computed tomography (CT) and magnetic resonance imaging (MRI) findings from actual patients [23,24,25]. The Anatomage Table™ was introduced to our institution in October 2021 for medical education, including lectures on physical examinations (Figure 1). These tools represent reliable solutions for improving medical student training, especially during the COVID-19 pandemic. However, there is a bit of a learning curve for familiarizing oneself with the use of the Anatomage Table, and it is a very expensive instrument, which raises the issue of its cost-effectiveness. Such issues should be further explored in the future [22].

## 3. Intraoperative Medical Education

High-precision visualization and documentation of the surgical field through advances in DX (digital imaging, WiFi internet connectivity, screen technology, and optics) is useful for surgical education of medical students and residents. In addition to conventional microscopes and endoscopes, exoscopes have been developed to enable the transmission and recording of highly accurate surgical information [26]. Furthermore, high-resolution videos captured with action cameras and 360° operative cameras, which provide an immersive and realistic experience, are relatively inexpensive and widely used in most areas of surgery, as well as various clinical education settings. The fields of application of 3D medical imaging and holograms are also diverse and promising, and these modalities are beginning to be used for education, surgical simulation, guidance, and navigation [3].

### 3.1. High-Resolution Videos Captured with Action Cameras and 360° Operative Cameras

The ideal camera for recording surgery is small, lightweight, comfortable, easy to use, able to depict the surgeon’s point of view, able to provide high-definition images and video, able to function for a long time on a single charge, inexpensive, and easily managed with regard to images and video [27]. Since such an ideal camera does not currently exist, in many cases, people cope by using different cameras for different purposes or by using combinations of multiple cameras.

High-resolution videos captured with an action camera from the surgeon’s point of view and 360° operative video have become more realistic and immersive through online live broadcasts of surgery. Action cameras that have been applied for surgical recording include the GoPro Hero Series (GoPro, Inc., San Mateo, CA, USA) [28,29], which can collect blur-free video images from the surgeon’s point of view. The GoPro is a small, lightweight, high-resolution action camera that has mainly been used to record action sports but that has recently been used in clinical education. When worn on a head mount, it has a 149° wide-angle field of view and can capture most of the visual information available to the wearer [27]. Furthermore, 360° operative video has been used not only to analyze surgical performance for medical education but also for orientation in new environments, team training, and formal multidisciplinary examinations, as it allows users to view images from all angles at the same time, providing an immersive and realistic experience [30]. We have been experimenting with the use of GoPro and 360° cameras to shoot surgical videos for spine surgery and lower-limb arthroplasty. The 360° camera is mounted on the handle of the operating light with a tripod to record all of the movements of the surgeon, assistants, anesthesiologists, and other staff from the center of the operating room to the entire room (Figure 2a–c), while the GoPro mounts on the surgical helmet worn by the surgeon capture the surgeon’s point of view (Figure 2d–f).

The videos from these two cameras are then edited and used for the education of students and residents. Watching a surgical video from the surgeon’s point of view provides a different experience from that of participating in surgery as an assistant, an experience that is strengthened further by watching with an HMD. In addition, using an HMD to watch the 360° operating room video from above the surgical field provides much more information than standing at the wall of the operating room. Thus, high-resolution videos may transform experts’ tacit knowledge into formal, shareable knowledge with the potential to improve the resident learning curve.

The video from the 360° camera can be trimmed to include only the part a student might like to watch. In this manner, we isolated the video including the surgeon, assistant, and scrub nurse in addition to the surgical field, and created a video that can be watched at the same time as the main video from the surgeon’s point of view. This video, created using consumer technology, is useful for the education of medical students, residents, and new staff, as it enables them to review not only the operation from the surgeon’s perspective but also the movements of each person (surgeon, assistant, anesthetist, and nurse) as the operation progresses. In addition, the use of high-resolution images captured by action cameras and 360° surgical cameras may also improve the effectiveness of medical education by providing medical students with hands-on experience from an early age.

### 3.2. Surgical Simulation/Guidance with 3D Holograms

Conventional 2D imaging techniques mainly include X-ray, CT, and MRI, but they require years of clinical experience to perform appropriately and a high degree of spatial imagination to be accurate. It is also difficult for medical students and residents to evaluate intraoperative images and understand surgical techniques because they cannot accurately grasp the 3D positioning of organs.

Case-specific 3D holograms may be a useful new educational tool for improving the competence of medical students and residents, as well as for educating patients [3,31]. In terms of teaching surgical anatomy, holograms developed with XR technology have made it easier for instructors to teach their material, as well as for students to understand it, suggesting a reduction in the gap concerning the perception of anatomical structures between teachers and students. Three-dimensional holograms of bones are particularly useful in spinal surgery, as they can be easily created using precise models. Furthermore, there are an increasing number of reports of case-specific 3D hologram-based surgical simulations/guidance [3,31,32,33,34,35,36,37,38,39,40,41,42].

Sugimoto et al. [33] developed a medical image VR/MR automatic application web service termed Holoeyes XR with the Holoeyes MD cloud system (Holoeyes, Inc., Tokyo, Japan), which stereoscopically visualizes 3D images as holograms. Case-specific images from CT or MRI or ultrasound data acquired from the imaging department are uploaded to the cloud and automatically and instantly converted and mounted onto the corresponding HMD system through the hospital network. This innovative service allows surgeons to use the technology without the inconvenience of converting medical images into 3D VR data using complex computer language. Holoeyes XR uses VR HMD to educate students and residents on surgical anatomy and to perform preoperative simulations of difficult cases. Holoeyes MD enables holographic-based navigation by superimposing holographic elements onto the actual patient’s superficial anatomy in real time on the operating table with MR HMD. Intraoperative holograms may represent the next generation of surgical support tools in terms of spatial awareness, shareability, and simplicity [3,31,32,33,34,35,36,37,38,39,40,41,42].

The intraoperative use of 3D holograms with high spatial awareness enhances the safety of surgery. Furthermore, these 3D holograms can be easily manipulated by gestures, allowing them to be viewed and moved without a monitor in any location, including the operating theater. We have utilized this system (Holoeyes XR) in cases of spinal tumor and deformity to perform hologram-based surgical simulation preoperatively (Figure 3a–c) and hologram-based surgical navigation and guidance intraoperatively with Holoeyes MD (Figure 3d). While holograms are still technically inadequate for use in strict intraoperative navigation, they can function as useful intraoperative guides to aid staff in making objective decisions, especially residents.

## 4. Postoperative Rehabilitation

To maximize the effectiveness of surgery, rehabilitation is as important as surgery itself. The novelty and immersion of XR have been proven to promote motivation, excitement, and task engagement, making virtual rehabilitation more effective than traditional rehabilitation [43]. Recently, VR technology has been introduced into a variety of clinical settings, including physical, vocational, cognitive, and psychological rehabilitation [44].

VR has been used as a distraction to relieve pain, known as ‘VR analgesia’, and it has been shown to have analgesic effects. The gamification of task-oriented treatment also makes rehabilitation fun and motivates patients to carry out repetitive tasks. Therefore, XR technology-based devices can improve analgesia, compliance, and the effectiveness of rehabilitation in spine medicine. Studies on the use of VR HMD in orthopedic rehabilitation have been carried out to assess range of motion and analgesia. Assessing the range of motion of the cervical spine is useful for pre- and postoperative assessments of the cervical spine and for checking the effectiveness of rehabilitation. The VR system allows for a convenient, noninvasive, harmless to the human body, and highly accurate assessment of the range of motion of the cervical spine [45,46]. It can be used to study the relationship between data measured with VR HMD and clinical imaging data, as well as for remote follow-up [45]. Gumaa et al. [47] summarized in their systematic review that the evidence concerning VR effectiveness in cases of chronic neck pain is promising. Some reports have shown that VR rehabilitation, such as that involving horseback riding and dodgeball, increased lumbar flexion and muscle mass. In addition, the VR technology system also helps patients undergo standardized rehabilitation at home without having to visit the hospital [48,49]. The application of VR technology to telemedicine in the field of rehabilitation has the potential to reduce the cost and time burden of rehabilitation for both doctors and patients. We have used the VR rehabilitation medical device mediVR (MediVR Co., Ltd., Osaka, Japan) for the rehabilitation of cerebellar ataxia and spinal disorders. The mediVR provides patients with a standardized, tailor-made, dual-task exercise rehabilitation program for body trunk balance using VR and 3D tracking technologies (HTC Vive^®^; HTC Corporation, New Taipei City, Taiwan) [45]. The advantages of mediVR were found to be as follows: (1) the patient is immersed in the VR space, which facilitates feed-forward learning of the target movement without interference from the external environment; (2) the patient is in a seated position, which is safe and carries no risk of falling; (3) the speed and frequency of the task can be accurately adjusted, which is highly reproducible and combines objectivity and convenience; (4) the difficulty level can be adjusted according to individual characteristics and highly sensitive, accurate, and diverse feedback can be easily created and presented; (5) the level of difficulty can be adjusted to suit individual characteristics, and highly sensitive, accurate, and diverse feedback can be easily created and presented [48,49]. No conclusions have yet been reached regarding the effectiveness of VR for spinal diseases. Higher-quality clinical studies are, thus, needed to reach more solid conclusions.

## 5. Limitations

This study was not a review but a perspective, describing future directions and personal opinions on medical education and rehabilitation in the field of spine surgery. Therefore, the lack of systematic methodology in this study made it impossible to obtain the highest level of evidence concerning conditions and techniques available at present. However, in the field of spinal surgery, there are no reports of the use of Anatomage, action cameras, or 360° cameras in education, and there are fewer reports on the use of hologram-based guidance than on VR-based rehabilitation and hologram-based simulation. In addition, previous reviews of XR techniques in spine surgery have not been meta-analyses or adequate systematic reviews because of heterogeneity in the study design, outcome measures, and variability. The use of XR technology is reported as an issue meriting future consideration in the field of medical education and rehabilitation in spine surgery.

## 6. Conclusions

During the COVID-19 pandemic, XR techniques in medical education and rehabilitation have been increasingly frequently adopted in the field of spinal surgery. DX using XR technology will support medical education and rehabilitation in order to maximize the effectiveness of patient care in the field of spinal surgery.

## Figures and Tables

**Figure 1 medicina-58-00508-f001:**
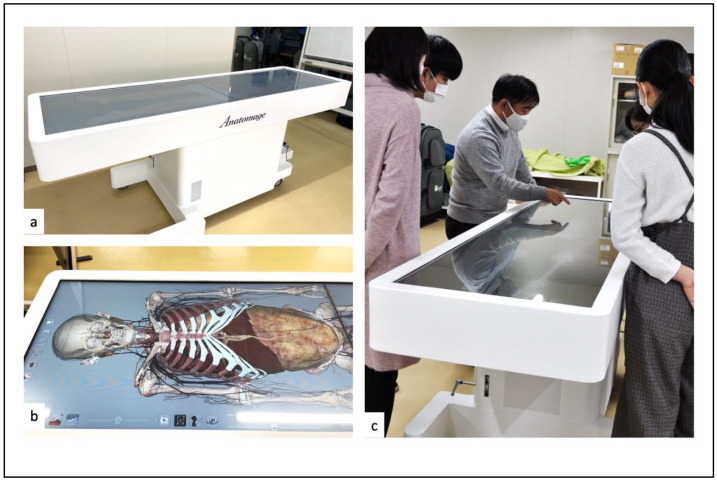
Anatomage Table™: (**a**) Anatomage Table in the horizontal position; (**b**) viewing anatomical images on the screen; (**c**) medical students working in a small group.

**Figure 2 medicina-58-00508-f002:**
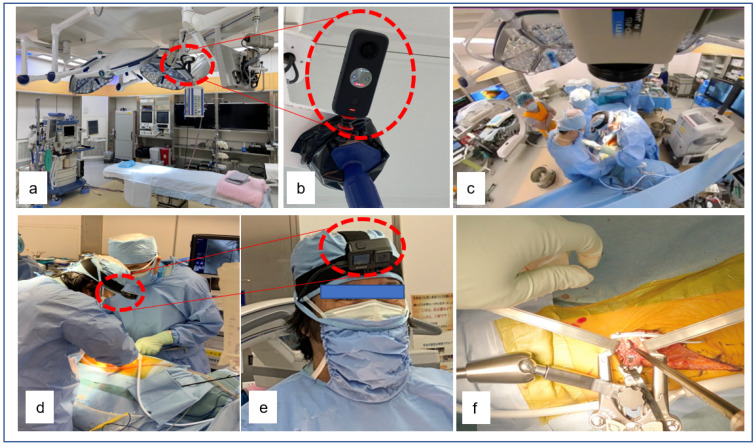
High-resolution videos: 360° cameras (**a**,**b**) and surgical videos for spine surgery (**c**). A video is taken by the GoPro (**d**,**e**) from the surgeon’s point of view (**f**).

**Figure 3 medicina-58-00508-f003:**
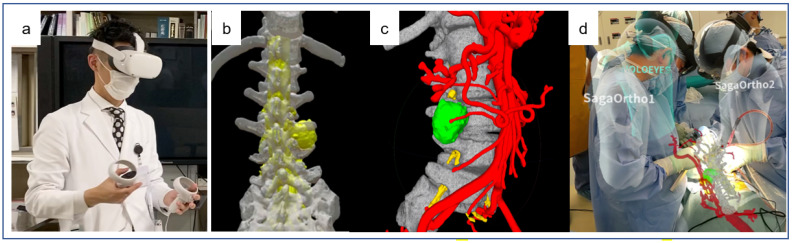
Hologram-based surgical simulation (**a**–**c**) and intraoperative guidance (**d**). (**a**) Resident doctor with a head-mounted display. (**b**,**c**) Relationship between schwannoma and spine and blood vessels in a virtual reality environment (Holoeyes XR). (**d**) Hologram-based surgical intraoperative guidance (Holoeyes MD).

## Data Availability

Not applicable.

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
