# Peer review of "Digital Transformation Will Change Medical Education and Rehabilitation in Spine Surgery"

_medicina, 2022, doi:10.3390/medicina58040508_

Round 1

Reviewer 1 Report

Interestingly, carefully consider these points:

  • Lines 76-80: "This study highlights the current medical education approaches implemented during... based rehabilitation, particularly in relation to MIST." Is this a review ? What is the aim of this paper?
  • If this paper is a review, please add a diagram flow in Methods section.
  • Line 132-135: "3.1. High-resolution videos captured with action cameras and 360° operative cameras" Authors should consider also the role of exoscope as new educational tool for young surgeon and medical students. doi: 10.3390/jcm11010223 
  • Lines 190-200: "3.2. Surgical simulation/guidance with 3D holograms" Please divide this part in surgical simulation (performed in laboratory, doi: 10.3390/ijerph18199955 ) and surgical guidance (performed in the operating room, doi: 10.3109/02688690903506093). Improve discussion, see refs.
  • The whole psper is very discursive, it should be more methodological and sequential, Improve. What this paper add new to the literature?
  • Lines 268-269: "No conclusions have yet been reached regarding the effectiveness of VR for spinal diseases" How do authors state that? include refs:  doi: 10.3390/ijerph19031442,  10.3389/fped.2021.760363
  • This paper has some limitations, such as that it does not consider single specialistic disease each one with its own characteristics, but considers surgery in its interaction. These limitations should be reported before the conclusion section.
  • What about spine surgery? Authors started with the title "Mini- 2
    mally Invasive Spine Therapy (MIST)", but in the conclusion section they did not report anything about MIST. This must be improved.

Author Response

Reviewers’ Comments to the Authors:

Reviewer 1

>Lines 76-80: "This study highlights the current medical education approaches implemented during... based rehabilitation, particularly in relation to MIST." Is this a review ?  What is the aim of this paper?

> If this paper is a review, please add a diagram flow in Methods section.

Author response: This study was not a Review but rather a Perspective, which generally presents examples of current developments in a field and focuses on the future direction of the field and the author's personal assessment (Ref. DOI: 10.3390/medicina57010081). In this study, the efforts to improve education for medical students, residents and patients using digital transformation (DX), which has flourished globally due to the COVID19 pandemic, especially in the field of spine surgery, are discussed as a Perspective. As Reviewer 1 pointed out, the purpose was unclear, so we have revised the description as follows (Line 75-77) and described the issue as a methodological one in the limitations section (Line 293-303):

This study presents an approach to pre-, intra- and post-operative medical education using the current DX that was implemented during the COVID-19 pandemic and will be utilised thereafter.

> Line 132-135: "3.1. High-resolution videos captured with action cameras and 360° operative cameras" Authors should consider also the role of exoscope as new educational tool for young surgeon and medical students. doi: 10.3390/jcm11010223 

Author response: As suggested, we have now added relevant text to Line 133-143 and described the utility of an exoscope, citing the Montemurro paper (Montemurro, N.; Scerrati, A.; Ricciardi, L.; Trevisi, G. The Exoscope in Neurosurgery: An Overview of the Current Literature of Intraoperative Use in Brain and Spine Surgery. Journal of Clinical Medicine. 2021, 11, 223.).

> Lines 190-200: "3.2. Surgical simulation/guidance with 3D holograms" Please divide this part in surgical simulation (performed in laboratory, doi: 10.3390/ijerph18199955 ) and surgical guidance (performed in the operating room, doi: 10.3109/02688690903506093). Improve discussion, see refs.

Author response: Unfortunately, this study is not a Review, and due to the limited number of words, we have not separated it into simulation and guidance. However, we did describe a report on surgical simulation, citing Montemurro et al. Furthermore, there are increasing reports of case-specific 3D hologram-based surgical simulations in the laboratory (Montemurro). We have also included an additional paper that you provided (Low et al.).

>The whole paper is very discursive, it should be more methodological and sequential, Improve.

Author response: As suggested, we revised the paragraph structure (Heading) and added text that explains the contents of this Perspective in greater detail.

Paragraph structure (Heading):

  1. Introduction
  2. Pre-operative medical education
  3. During-operative medical education
  4. Post-operative rehabilitation
  5. Limitation
  6. Conclusion

>What this paper add new to the literature?

Author response: To maximize the effectiveness of patient care, the educational needs of medical students, residents, and patient rehabilitation can be enhanced by digital transformation (DX) using XR technology. In the field of spinal surgery, there are no reports of the use of ANATOMAGE, action cameras or 360° cameras in education, and there are fewer reports on the use of holo-gram-based guidance than on VR-based rehabilitation and hologram-based simulation. The use of XR technology was reported as an issue meriting future consideration in the field of medical education and rehabilitation in spine surgery.

>Lines 268-269: "No conclusions have yet been reached regarding the effectiveness of VR for spinal diseases" How do authors state that? include refs:  doi: 10.3390/ijerph19031442,  10.3389/fped.2021.760363

Author response: As suggested, we have corrected VR for spinal diseases to VR-based rehabilitation for spinal diseases.

>This paper has some limitations, such as that it does not consider single specialistic disease each one with its own characteristics, but considers surgery in its interaction. These limitations should be reported before the conclusion section.

Author response: As you point out, it is a Perspective, not a Review, and we have described the methodological shortcomings in the Limitations section (Line 293-303).

>What about spine surgery? Authors started with the title "Minimally Invasive Spine Therapy (MIST)", but in the conclusion section they did not report anything about MIST. This must be improved.

Author response: Since the focus on this study was on education and rehabilitation in spine surgery rather than spine surgery itself, we have revised the title and conclusions as follows:

Title: Digital Transformation will change Medical Education and rehabilitation in Spine Surgery

Conclusions

During and after the COVID-19 epidemic, XR techniques in medical education

and rehabilitation have been increasingly adopted in the field of spinal surgery.

Digital transformation (DX) using XR technology will change medical education and rehabilitation to maximise the effectiveness of patient care in the field of spinal surgery.

Reviewer 2 Report

The manuscript “Digital Transformation of Medical Education related to Minimally Invasive Spine Therapy (MIST)” by Tadatsugu Morimoto et al. aimed to highlights the current medical education approaches implemented during the COVID-19 pandemic and likely to be used beyond, using digital technologies such as 3D virtual dissections, high-resolution surgical videos captured with action cameras and 360° operative cameras, 3D hologram-based surgical simulation and guidance and VR- based rehabilitation, particularly in relation to MIST.

The manuscript is an overview of known modern educational techniques.
For the features of a scientific article, it would be worthwhile to demonstrate, for example, changes in education in the numbers of students using these techniques before and during the pandemic.

Below are my comments and remarks regarding the manuscript:

1. There is no method section describing the method of data collection / text elaboration
2. What is new in manuscript?
3. Not specified particular surgical techniques / diseases
4. Failure to describe the limitations these should be reported before the summary section.
5. The title is misleading about spine surgery

Author Response

Reviewers’ Comments to the Authors:

Reviewer2

Below are my comments and remarks regarding the manuscript:

> 1. There is no method section describing the method of data collection / text elaboration

Author: This study was not a Review but rather a Perspective, which generally presents examples of current developments in a field and focuses on the future direction of the field and the author's personal assessment (Ref. DOI: 10.3390/medicina57010081). In this study, the efforts to improve education for medical students, residents and patients using digital transformation (DX), which has flourished globally due to the COVID19 pandemic, especially in the field of spine surgery, are discussed as a Perspective.

> 2. What is new in manuscript?

Author: In the field of spinal surgery, there are no reports of the use of ANATOMAGE, action cameras or 360° cameras in education, and there are fewer reports on the use of holo-gram-based guidance than on VR-based rehabilitation and hologram-based simulation. The use of XR technology was reported as an issue meriting future consideration in the field of medical education and rehabilitation in spine surgery in this manuscript as a Perspective.

>3. Not specified particular surgical techniques / diseases

Author:

The study focused on XR techniques in education and rehabilitation in spine surgery rather than specific surgical techniques / diseases.

>4. Failure to describe the limitations these should be reported before the summary section.

Author: We have now described the Perspective's approach to data collection as a limitation.

> 5. The title is misleading about spine surgery

Author response: Since the focus of the present study was education or rehabilitation in spine surgery rather than spine surgery itself, we have revised the title as follows:

Title: Digital Transformation will change Medical Education and rehabilitation in Spine Surgery

Round 2

Reviewer 1 Report

Authors solved all my criticisms.

Reviewer 2 Report

I have no more comments.